# Microstructure and Texture Evolution of Mg-Gd-Y-Zr Alloy during Reciprocating Upsetting-Extrusion

**DOI:** 10.3390/ma13214932

**Published:** 2020-11-03

**Authors:** Guoqin Wu, Jianmin Yu, Leichen Jia, Wenlong Xu, Beibei Dong, Zhimin Zhang, Biying Hao

**Affiliations:** 1School of Material Science and Engineering, North University of China, Taiyuan 030051, China; wuguoqin5656@hotmail.com (G.W.); jlc226688@hotmail.com (L.J.); wenlong0610@hotmail.com (W.X.); dongbb1111@163.com (B.D.); ZZMNUC@163.com (Z.Z.); 2Xi’an Modern Chemistry Research Institute, Xi’an 710065, China; yujianmin@hotmail.com

**Keywords:** Mg alloys, texture evolution, DRX mechanism, SPD, microstructure evolution

## Abstract

Reciprocating Upsetting-Extrusion (RUE) deformation process can significantly refine the grains size and weaken the basal plane texture by applying a large cumulative strain to the alloy, which is of great significance to weaken the anisotropy of magnesium (Mg) alloys and increase the application range. In this paper, the Mg-8.27Gd-3.18Y-0.43Zr (wt %) alloy was subjected to isothermal multi-passes RUE. The microstructure and texture evolution, crystal orientation-dependent deformation mechanism of the alloy after deformation were investigated. The results clearly show that with the increase of RUE process, the grains are significantly refined through continuous dynamic recrystallization (CDRX) and discontinuous dynamic recrystallization (DDRX) mechanisms, the uniformity of the microstructure is improved, and the texture intensity is reduced. At the same time, a large number of particle phases are dynamically precipitated during the deformation process, promoting grain refinement by the particle-stimulated nucleation (PSN) mechanism. The typical [10-10] fiber texture is produced after one pass due to the basal plane of the deformed grains with a relatively high proportion is gradually parallel to the ED during extrusion process. However, the texture concentration is reduced compared with the traditional extrusion deformation, indicating that the upsetting deformation has a certain delay effect on the subsequent extrusion texture generation. After three or four passes deformation, the grain orientation is randomized due to the continuous progress of the dynamic recrystallization process.

## 1. Introduction

Magnesium (Mg) alloys as the first choice for lightweight materials have attracted wide attention in the aerospace and military fields due to their high specific strength, specific stiffness, excellent thermal conductivity [1,2,3]. However, the poor mechanical properties and strength at room temperature (RT) greatly limit the practical application of wrought Mg alloys, which was related to the hexagonal close-packed (HCP) structure of Mg alloys [4,5]. Based on the HCP lattice structure, only the basal slip [0001] <11-20> can be activated when the Mg alloy is deformed at RT, and the multi-slip system is only activated when the deformation temperature is higher than 250 °C [6]. In addition, the generation of strong texture under some conventional plastic deformation, such as extrusion or roll, causes anisotropy of mechanical properties and further limits the application of Mg alloy [7,8,9]. Therefore, improving formability and optimizing texture have become important research directions for extending the application fields of Mg alloy.

Severe plastic deformations (SPDs) can effectively refine grains and weaken texture through multiple-passes repeated deformation of a certain volume of metal to obtain a large cumulative strain [10], thus attracting a large number of researchers [11,12,13,14]. Dong [11] et al. applied the homogenized Mg-13Gd-4Y–2Zn-0.5Zr to multi-directional forging (MDF) process and found that the maximum failure elongation (FE) was 18.1% after 4 passes by refinement strengthening mechanism. Krzysztof [12] research shows that the microstructure of ZE41A Mg alloy is significantly refined after four passes of equal-channel angular pressing (ECAP), and the strength is increased to twice that of the as-cast state. Among the many SPDs technologies, the RUE technique combining upsetting and extrusion deformation together provides a good plastic processing method for difficult-to-deform Mg alloys and can be used for pre-forming of huge parts and has low equipment requirements. At the same time, the compression and extrusion stress states are alternately loaded during the RUE process, which is beneficial to promote the continuous transformation of the orientation of the basal plane, and finally achieve the texture weakening. Dong [15] et al. carried out CEE of Mg-4Gd alloy and found that the texture intensity increased with the increase of deformation passes, and the grains rotated violently during the deformation process. Zhang [2] et al. investigated the microstructure and texture of GWZK124 alloy under RUE deformation, and the results showed that the particles and second phases induced dynamic recrystallization (DRX), which can be explained by the particle-stimulated nucleation (PSN) mechanism. However, the relationship between grain refinement and texture weakening in the RUE deformation process has not been well reported, which has a good guiding significance for improving the RUE deformation process.

In addition to weakening the texture and improving the forming performance of the alloy from the perspective of the deformation method, another important way is to add rare earth elements for alloying. For instance, the addition of Zr element could refine the grains size and the Gd element was beneficial to the formation of precipitation phase with strong thermal stability [16,17,18,19]. However, the researches on this alloy are currently mainly focused on conventional deformation such as extrusion, compression, and heat treatment. For example, Zhang et al. [20] studied the relationship between flow stress of Mg-8Gd-3Y-0.6Zr and deformation temperature and deformation rate; LI et al. [21] had shown that the main deformation mechanism of GW83K alloy is grain boundary slip during compression deformation at 400 °C and 10−4 s−1. Although previous research can improve the mechanical properties of the experimental alloys to a certain extent, the poor plastic processing methods still cannot meet the industry’s requirements for alloy properties.

In this paper, the Mg-8.27Gd-3.18Y-0.43 Zr (wt %) alloy was subjected to isothermal multi-passes RUE process. The microstructure and texture evolution during RUE process were investigated. Further, the relationship between texture evolution and dynamic recrystallization behavior were also examined in detail. Understanding the relationship between microstructure evolution and texture change during the deformation process is conducive to obtaining a reasonable texture type and intensity by control the deformation mode or deformation parameters.

## 2. Materials and Methods

The Mg-8.27Gd-3.18Y-0.43 Zr (wt %) alloy used in this research is a cylindrical bar with 50 mm diameter and 220 mm height turned from the cast rod. Before RUE deformation, the material was homogenized at 520 °C for 16 h to eliminate dendrite segregation that occurs during the casting process, followed by air cooling. The RUE processing was carried out by pushing a specimen into the cavity with the diameter of D for upsetting firstly and then extruded it with the diameter of d, as shown in Figure 1a. As can be seen, the samples can be restored to the initial processing state after each pass. The billet diameter D and d after upsetting and extrusion are 70 mm and 50 mm, respectively, as shown in Figure 1b. It can be calculated that the extrusion ratio is 1.96. Figure 1c shows a schematic diagram of the working part of the extrusion die. The experiment was conducted on a 4-THP61-630 pressing machine with 630 t capacity. The samples obtained from the former pass were reheated to 420 °C for 2 h before the subsequent pass, and one upsetting and one extruding process were regarded as one pass, the whole experiment including four passes. The true strain of each pass was 1.35 according to equation:E = 4nln(D/d)(1)
where n and ε represent deformation pass and cumulative strain, respectively. Samples with different RUE passes were designated as the RUE1, RUE2, RUE3, and RUE4, respectively.

The microstructure of RUE samples is characterized by optical microscope (OM, DM2500M, Leica Microsystems, Wetzlar, Germany) and scanning electron microscope (SEM, SU5000, Hitachi, Tokyo, Japan) under an accelerated voltage of 20 kV. Specimens for OM measurement are mechanically polished and chemically etched using a solution with 1 g picric acid, 2 mL acetic acid, 2 mL distilled water and 14 mL alcohol. X-ray diffraction (XRD) was used for phase identification during RUE process. Furthermore, the SEM equipped with electron back-scattered diffraction (EBSD) was used to observe the evolution of grain orientation. TSL OIM analysis version 7.3 (EDAX Inc, Mahwah, NJ, USA) was used to quantify the details on the grain misorientation angles, grain size, micro-texture, kernel average misorientation, and Schmidt Factor.

## 3. Results

### 3.1. Evolution of Microstructure

Figure 2 demonstrates the optical and SEM microstructure of sample before RUE. As shown in Figure 2a, the as-cast sample consists of α-Mg and coarse eutectic phases. S.S.A. et al. [22] have identified them as β-phases (Mg24(Gd, Y)5). After homogenized at 520 °C for 16 h, the eutectic β-phase almost completely dissolved into the matrix (Figure 2b). However, there are still a few cuboid shaped particles remaining, as shown in Figure 2b, marked by red arrow. Gao et al. [23] have identified them as Mg5(Gd, Y) phases with FCC crystal structure.

The microstructure evolution of the samples is shown in Figure 3 and Figure 4, which represent the OM and SEM-BSE (back-scattered electron) diagram of the samples RUE1, RUE2, RUE3, and RUE4, respectively. It can be clearly seen that with the RUE passes increasing, more and more fine DRXed grains, as indicated by the blue dashed line, are produced along the grain boundaries, resulting in continuous grain refinement. At the same time, the degree of uniformity of microstructure is also increased with the increase of RUE passes due to the generation of DRXed grains. With the deformation passes up to 4, the microstructure of the sample is almost all equiaxed crystals obtained by DRX, as shown in Figure 3d.

Figure 4 shows that the number of precipitated phases marked by the red arrow increases significantly from 1 to 2 passes, and the dispersion degree also increases. In the 1 RUEed sample (Figure 3a), only a small amount of particle phases is found at the DRXed grain boundaries (GB), as indicated by blue arrow, but not in the unDRXed regions (coarse deformation grain indicated by yellow arrow). The discontinuous distribution of particle phases determines that the alloy is a bimodal microstructure composed of DRXed grains and unDRXed grains. After second passes (Figure 3b), the particle phases precipitate at the grain boundary and inside the DRX grain at the same time, as shown in the red dashed arrow, and the number of precipitated phases increased, but the distribution is still not continuous. After 3 and 4 passes (Figure 4c,d), the number of particle phases are significantly increased and diffusely distribute, which may be due to the repeated axial and radial flow of the metal during SPD. As illustrated in Figure 5, the RUEed samples are composed of α-Mg matrix and Mg5 (Gd, Y). Xiao et al. [24] also found a large amount of Mg5(Gd, Y) phases, which have an FCC crystal structure with a lattice constant a = 2.23 nm, after hot compression of Mg–Gd–Y–Zr alloy.

Study [24] had shown that the dynamic precipitation of equilibrium β phase, Mg5(Gd, Y) phase, only occurred in the range of 300–450 °C during compression deformation and mainly determined by the solid solubility of the solute atoms. In addition, it is found that the Mg5(Gd, Y) phases easily appear at the DRXed GBs, as shown in Figure 3b, which is owing to the strain induction precipitation. A large number of dislocations accumulate at the GBs causing serious distortion of the crystal lattice, which provides a channel for atomic diffusion and the required energy [25]. During the RUE process, the strain accumulation increases with the increase of passes, then the driving force provided by deformation energy increases to accelerate the precipitation of the particle phases. Therefore, the number of precipitated phases rises continuously and evenly distributed throughout the sample.

In order to further understanding the deformation mechanism and lattice orientation texture during RUE process, the microstructures after RUE were further investigated using EBSD. Figure 6 shows the grain orientation distribution of the microstructure, DRXed grains separately, grain size and misorientation angle distribution of the RUEed samples in different passes. The grain orientation spread (GOS) defined by calculating the average misorientation be- tween all pixel point within each grain was applied to count the dynamic recrystallization (DRX) fraction of tested samples [26]. In this work, the DRXed grain was identified by the GOS value smaller than 2°, while the unDRXed grain greater than 2°. Besides, the different colors represent different grain orientations, as shown in the triangle card in Figure 6a. It can be clearly seen that the proportion of DRXed grains increases with the RUE passes increase. The bimodal structure composed of fine DRXed grains and coarsely deformed grains (Figure 6a) gradually transforms to microstructure with equiaxed fine grains (Figure 6d).

Figure 6a shows that the original grains are elongated along extrusion direction (ED) after 1 pass. When the cumulative strain is low, only a small amount of DRXed grains are produced along the GBs of the coarsely deformed grains, owing to dislocations accumulating, and merge easily occurs at the GBs, resulting in extremely uneven microstructure. As the cumulative strain increases to 2.7 (Figure 6b), the dislocation proliferation generated at the Mg matrix and GBs as the main deformation mechanism causes the production of more DRX grains, and the grain size is obviously refined compared with 1 pass. After 3 and 4 passes, a large number of DRXed grains are generated by consuming the large deformed grains, the microstructure becomes more homogeneous, and the average grain size is reduced from 48.8 μm to 4.5 μm.

Figure 6e–h clearly shows that the proportion of DRX grains increase during RUE process, and the average grain size of DRXed gradually decrease from 14.1 μm to 4.3 μm. Mamoru M et al. [27] proved that the grain size of DRXed are affected by the parameters of Zener–Hollomon. To put it simply, when the strain rate and deformation temperature change, the average grain size of DRXed will also change. What is interesting is that the deformation temperature and strain rate does not change in this study, but the grain size of DRXed decreases. Xiao.et al. [24] have shown that the preferentially precipitated particle phase Mg5(Gd, Y) caused [28] by segregation of Gd and Y rare-earth elements at GBs have high thermal stability and have a pinning effect on the grain boundary slip (GBS) during deformation process. Therefore, the migration of the GBS of the DRXed grains can be suppressed during thermal deformation process and heating process between passes. At the same time, the particle phase also can provide nucleation sites for the formation of DRXed grains and promote the DRX process, which is called particle excitation nucleation (PSN) mechanism [29,30]. This is also one of the reasons why the proportion of DRX grains continues to rise during deformation process.

Figure 6i–l shows that the distribution range of grain size gradually decreases with the increase of RUE passes. The distribution range of grain size ranges from 2 μm–120 μm for one pass to 2 μm–15 μm for four passes. The standard deviation value decreased from 29.7 of 1 pass to 1.5 of 4 passes, quantitatively indicating that the uniformity of the microstructure is significantly improved with the RUE passes increase, and the metal flow is sufficient. It is worth noting that the grain size does not decrease linearly with the increase of deformation passes; that is to say, the refinement ability gradually decreases during RUE process. This may be related to the dislocation density available for consumption. In the process of continuous deformation, strain hardening causes dislocations in grains of different orientations. Under the action of stress, the dislocations slide along the basal or non-basal surface, and when they slip to the initial grain boundary, dislocations are generated. When the accumulation reaches a certain level, the dislocations are rearranged and merged (dynamic recovery), resulting in dislocation cells and sub-grain boundaries. The sub-grain boundary can increase its misorientation by continuously absorbing lattice dislocations and transforms into HAGB, completing the CDRX process. Therefore, the percentage of dislocation propagation will decrease with the proportion of DRX grain increases. When the dislocation density is lower than the nucleation requirement, the degree of grain refinement by DRX will decrease.

Figure 6m shows that the fraction of low-angle grain boundary (LAGB) in 1RUEed sample increases to 65%, indicating the formation of numerous sub-grain boundaries and the operation of CDRX. With the increase of passes, the proportion of LAGBs decreases continuously. After 2 passes, the fraction of LAGBs declined to 35%, and the fraction of high-angle grain boundaries (HAGBs) increases to 60%, which can be ascribed to the almost complete DRX process at higher accumulative. The average misorientation angle also increased from 23.1 to 37.6. The dislocation density increases with RUE passes increase, then gradually transforming to sub-grains or LAGBs. As the cumulative strain continues to increase, the previously formed sub-grains or LAGBs will absorb the new dislocations and transform into HAGBs, which is a typical CDRX process. After 3 or 4 passes deformation, the average misorientation angle increases slightly, but the microstructure is still significantly refined, indicating that another DRX mechanism different from CDRX happens. From the previous description, it can be known that numerous particle precipitation phases are produced in RUE3 and RUE4 samples, which may contribute to the DDRX process through the PSN mechanism [30].

### 3.2. Texture Evolution

When the alloy is continuously deformed, the texture will change significantly due to the formation of preferential orientation, especially in the Mg alloy with less slip system [31,32,33]. Figure 7 represents the basal pole figure (PF) and the corresponding inverse pole figure (IPF) of the RUEed alloy under different passes. when the cumulative strain up to 1.35, the typical extrusion basal plane texture is produced accompanied by a certain angle deviation, as shown in Figure 7a. The corresponding IPF (Figure 7e) shows a strengthened [10-10] fiber texture accompanied with other weaker texture components. Aidin et al. [8] reported that the addition of rare earth elements would hinder the formation of sharp fiber texture in Mg alloys to form this weaker texture component. In addition, it is worth noting that the maximum pole density region in the IPF diagram deviates from [10-10] poles by about 30 degrees, which is more dispersed than the typical extrusion texture.

After cumulative strain up to 2.7, the (0001) pole figure (Figure 7b) shows a basal texture type does not changed significantly than 1 pass, but the maximum pole density area is deflected toward two poles, indicating that the concentration of the grain orientation is reduced, that is, the texture intensity reduced. The IPF (Figure 7f) shows a strong peak concentrated at [10-10], and there is a tendency to shift to the double fiber texture, compared with Figure 7e. After 3 passes, the sample (Figure 7c) exhibits relatively random basal plane orientation accompanied with obvious [10-10]–[2-1-10] double fiber texture (Figure 7g). After 4 passes, the grain orientation is further dispersed (Figure 7d), but the texture strength increases slightly accompanied with the formation of rare earth texture [8]. The dispersion of grain orientation after three or four passes is closely related to the increase in the proportion of DRXed grains, which will be clarified in Section 4. In general, with the deformation passes increases, the fiber texture produced after one deformation gradually disappears, and the texture intensity also weakens. It is worth noting that the RUE is an axisymmetric metal forming technique, the pole density area in the pole figure should be distributed symmetrically around the RUE axis (ED), but the deformation texture of 3 or 4 passes does not show symmetry, which may be due to the limited observation range of EBSD.

Figure 8 shows the IPF coloring maps of DRXed and unDRXed grains after one-pass and three-passes deformation. The crystallographic orientation of unDRXed grains was separately highlighted in blue and presented in corresponding PFs and IPFs. Obviously, in Figure 8c,e, although the DRXed grains appear in dispersed orientations, caused the texture to be weakened to a certain extent, lots of unDRXed grains with [10-10] fiber orientation result in the texture feature after one pass, which is highly consistent with the texture in Figure 7a,b. In the three-passes deformed sample (Figure 8e,g,h), it can be clearly seen that the large number of DRXed grains causes the grain orientation to be dispersed, and the texture is obviously weakened. Therefore, in the continuous deformation process of RUE, the formation of texture is jointly determined by deformed grains and DRXed grains, depending on the proportion of them.

## 4. Discussion

In order to further reveal the influence of different dynamic recrystallization mechanisms on orientation evolution in unDRXed regions and DRXed regions during RUE process, typical unDRXed regions R1 and DRXed regions R2 were selected from 1, 2, passes, respectively, and analyzed in detail. The results are presented in Figure 9 and Figure 10.

### 4.1. The Effect of unDRXed Region on Grain Orientation

It can be found from Figure 9 that there are several different orientations of unDRXed grains: (1) non-basal orientation such as G1 and (2) [10-10] fiber orientation such as G2. In the process of extrusion, the alloy is subjected to triaxial compressive stress in the central area without the influence of friction, where the radial compressive stress ($\sigma_{r}$) is equal to the circumferential compressive stress ($\sigma_{\theta }$), and both them are greater than the axial compressive stress ($\sigma_{1}$). Therefore, the combined action of radial and circumferential compressive stresses can be considered equivalent to compressive deformation, while the axial stress causes the metal to flow axially. Therefore, the crystal grains will be compressed along the radial direction during the extrusion deformation, causing them to be elongated along the ED. At the same time, the c-axis of the grains is deflected radially, and a large number of crystal grains whose basal plane is parallel to the ED are generated. However, during the RUE deformation process, the upsetting deformation makes part of the grains hard-oriented in the subsequent extrusion deformation process, and the basal slip system is difficult to activate, which may form deformed grains G1 with non-basal orientation. This reflects that the RUE has a delayed effect on the formation of the extrusion texture.

Due to the different initial orientations, the order of the grains forming the basal plane orientation with the extrusion deformation is different. Some grains with preferential basal plane orientation continue to deform due to stress in the subsequent deformation, and gradually change the grain orientation. Figure 9b shows a typical unDRXed grain G2 with basal plane orientation that is going to transform to DRXed grain. The point-to-origin misorientation along AB in G2 increases from 0 to 19, indicating the continuous change of orientation occurring inside G2, and the different colored regions also illustrate the change of orientation. It can be found the regions of different colors have the same basal plane orientation (0001)‖ED in the corresponding PF (Figure 9e), which are highlighting the same color with grain orientation. In addition, the inverse pole diagram (Figure 10f) shows the gradual change of orientation from <10-10>‖ED to <2-1-10>‖ED inside G2. The schematic diagram of the three-dimensional lattice distributed along the AB line also shows that the Mg unit cells are distributed along the basal plane parallel to the ED and gradually rotate around the c axis, which is consistent with the grain orientation formed by extrusion.

The LAGBs represented by the white line in G2 are located at the position separating the region of different orientation. As the amount of strain increases, the previously formed sub-grains or LAGBs will absorb the new dislocations and transform into HAGBs, completing the typical CDRX process. In order to further understand the deformation mechanism of this transformation, Figure 9f,g calculates the SF of G2 in different slip systems, and finds that the SF of prismatic slip is higher than basal slip, therefore, prism slip is likely to be the main deformation mechanism in some grains, which have the same basal orientation as G2. When prism slip dominates the formation of grain with c-axis close to TD, the crystal rotates 30 degrees around the c-axis without relative rotation on the c-axis, thus <10-10>∥ED is converted to <2-1-10>∥ED, as shown in Figure 9h [24]. This is consistent with the lattice rotation in G2. The CDRX process formed by non-basal slip causes the formation of the lattice orientation with the c-axis continuously distributed in the TD and ND planes.

### 4.2. DRX Behavior and Its Influence on Grain Orientation

During the subsequent upsetting deformation of the alloy after one pass, the c-axis of unDRXed grains to continue to deflect to ED, but initial grains (RUE1) orientation caused by extrusion weakens this deflection, which makes the angle between the c-axis of unDRXed grains and the equatorial axis decreases after extrusion during 2 passes, as shown in Figure 10c. However, the texture intensity of 2 passes is lower than 1 pass, which is closely related to the decrease in the proportion of unDRXed grains after 2 passes. This also shows that RUE has a delayed effect on the formation of extrusion texture.

As the proportion of DRXed grains continues to rise, the grain orientation is gradually randomized after 3 passes. It can be seen from the previous description that CDRX process only rotates the crystal around the c-axis without changing basal plane orientation, so another DRX mechanism different from CDRX may have occurred during the later deformation. Chen et al. [34] performed CEC deformation on AZ31 alloy, indicating that the DDRX mechanism is also one of the important mechanisms of grain refinement. Study [8] shows that DDRX process is caused by unbalanced strain energy and the grain boundaries of undergoing DDRX process are jagged like Figure 11a, which can be seen in Figure 6c. Sitdikov et al. [35] revealed that DDRX includes following steps: Firstly, as the cumulative strain increases, a large number of dislocation slips are generated and are blocked by HAGB, but the dislocation density on the other side of the grain boundary is lower, resulting in the HAGB to protrude and nucleate to adjacent grains. As the grain boundary slip occurring, the formation of a strong strain gradient causes the grain boundaries to emit dislocations into the grains. These dislocations interact with the basal plane dislocations to cut off the protruding parts of the sub-crystals, thus forming new grains with completely different orientations from the parent crystals, as shown in Figure 11b. However, the strain energy required by DDRX is relatively high, and therefore, DDRX mechanism only occurs when the cumulative strain reaches a certain values.

The texture intensity is further weakened accompanied by the dispersion of grain orientation after 4 passes. In addition to the increase in the proportion of DRXed grains formed by DDRX, it is also closely related to the deformation mode. With continuous grain refinement during RUE process, the texture formed by the single upsetting or extrusion deformation becomes more restrictive to each other, thus restricting the formation of a strong texture. It should be noted that the addition of rare earth elements also has an important effect on the texture weakening and the evolution of texture types. Compared with the texture obtained by repetitive extrusion deformation of pure Mg by Dong et al. [15], grain orientation of Mg-Gd-Y-Zr alloys is more discrete after RUE deformation. This may be due to the addition of rare earth elements. Studies [7,8] showed that the addition of rare earth elements can promote the uniformity of microstructure and reduce the energy of the grain boundaries, thereby increasing the orientation of the grain boundaries and restricting the specific grain orientation behavior of Mg alloys; rare earth elements also can reduce the CRSS of <c+ a> slip, thereby promoting DRX process by increasing the density of dislocation, then ultimately reducing the texture intensity. Mackenzie et al. [36] also proposed that PSN is an active mechanism contributing to the development of a weaker texture during extrusion of WE43 alloy. These mechanisms work together to form a dispersed grain orientation.

## 5. Conclusions

In this paper, Mg-Gd-Y-Zr alloy is used for multi-passes isothermal RUE deformation. The microstructure and texture evolution of materials during RUE process are studied. In addition, the relationship between the grain refinement mechanism and texture weakening is investigated. The main conclusions are summarized as follows:The RUE deformation has a significant effect on the grain refinement of the Mg alloy. The fraction of DRXed grains is significantly improved and homogeneous microstructure with statistically average grain size of 7.6 μm was obtained.Mg_5_(Gd-Y) phase is dynamically precipitated during RUE process. With the increase of cumulative strain, the number of particle phases also increases, which is beneficial to promote the generation of DRXed grains through the PSN mechanism and accelerate the process of microstructure refinement. In addition, the precipitated particle phase greatly hinders the migration of GBs through the pinning effect, thereby hindering the growth of DRXed grains.RUE deformation can effectively weaken the basal plane texture through CDRX and DDRX mechanisms. In this deformation process, the upsetting has a certain effect on the deflection of the c-axis ED in the subsequent extrusion deformation, so the grain orientation is gradually dispersed with the increase of passes. At the same time, the rare earth elements are added to the alloy. It also has a certain effect on texture weakening.

## Figures and Tables

**Figure 1 materials-13-04932-f001:**
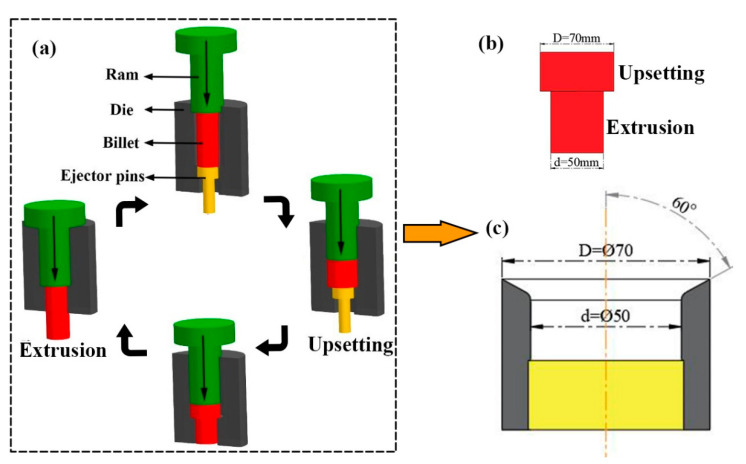
The schematic diagram of Reciprocating Upsetting-Extrusion (RUE) deformation (**a**), sample schematic diagram after upsetting and extrusion (**b**), and the dimension of the extrusion die (**c**).

**Figure 2 materials-13-04932-f002:**
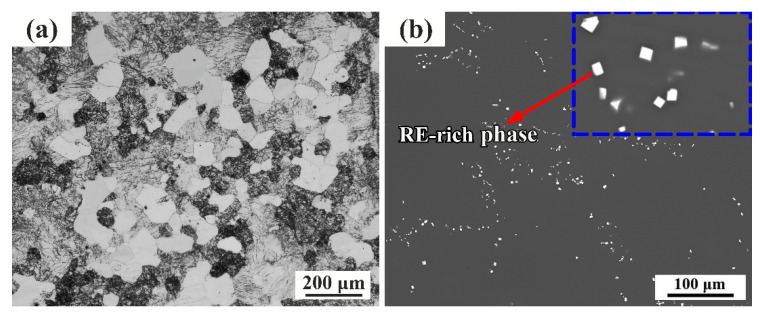
The optical microscope (OM) microstructure of as-cast alloy (**a**) and back-scattered electron (BSE) map of sample after homogenization treatment (**b**).

**Figure 3 materials-13-04932-f003:**
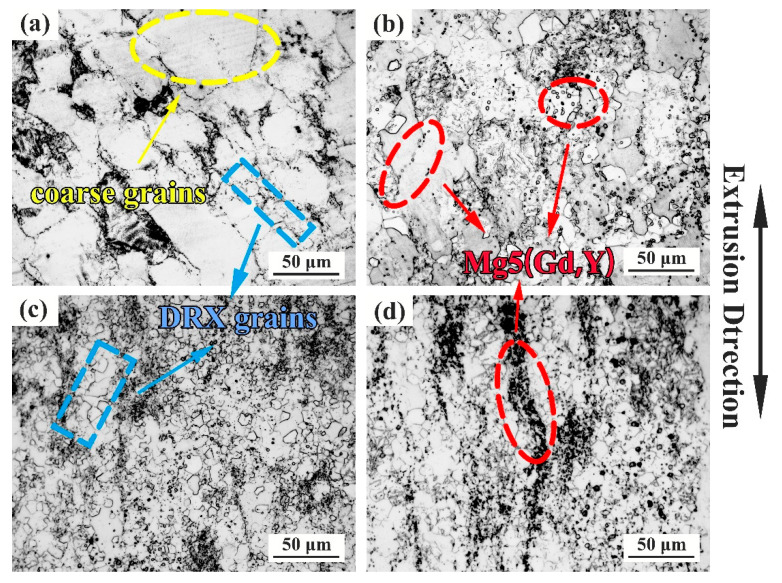
The OM microstructure of the samples RUEed after different passes. (**a**) 1 pass; (**b**) 2 passes; (**c**) 3 passes; (**d**) 4 passes.

**Figure 4 materials-13-04932-f004:**
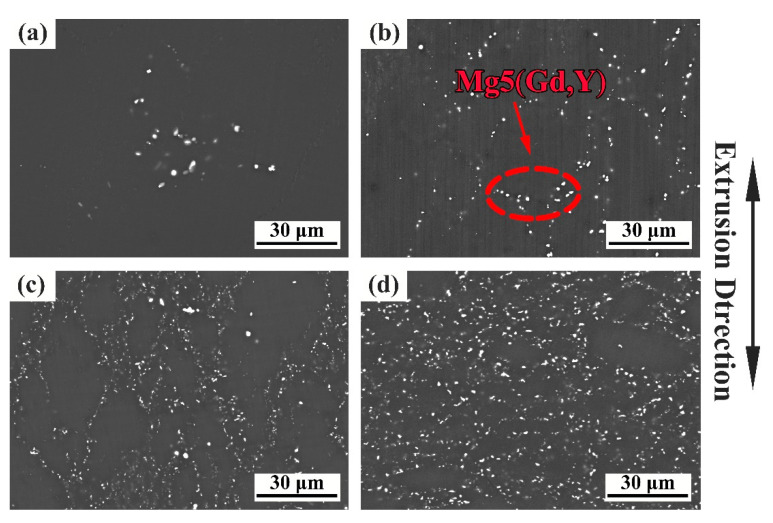
The back-scattered electron (BSE) of the samples RUEed after different passes. (**a**) 1 pass; (**b**) 2 passes; (**c**) 3 passes; (**d**) 4 passes.

**Figure 5 materials-13-04932-f005:**
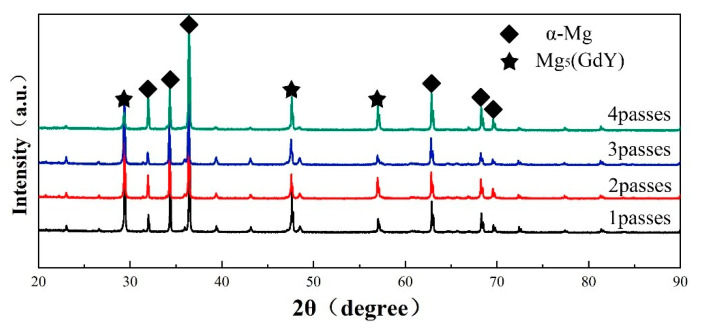
XRD patterns of alloys under different passes.

**Figure 6 materials-13-04932-f006:**
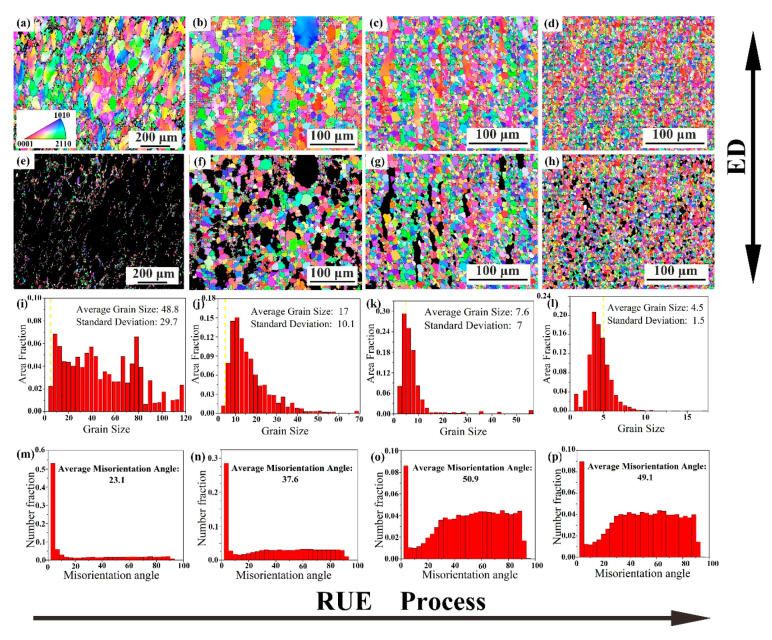
Grain orientation map (**a**–**d**); DRX grain orientation map (**e**–**h**); Grain size (**i**–**l**) and Misorientation distribution map (**m**–**p**). (SEM-EBSD) of the sample deformed in different passes at 420 °C: (**a**,**c**,**i**,**m**) 1 pass; (**b**,**f**,**j**,**n**) 2 passes; (**c**,**g**,**k**,**o**) 3 passes and (**d**,**h**,**i**,**p**) 4 passes.

**Figure 7 materials-13-04932-f007:**
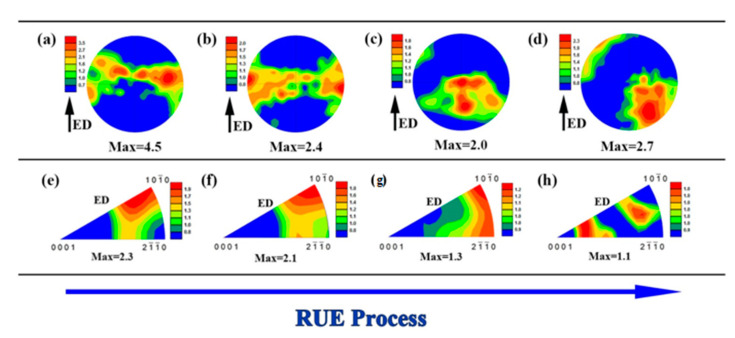
Texture development during RUE:(0001) pole figures (**a**–**d**) and corresponding inverse pole figures (**e**–**h**), 1 pass (**a**,**e**), 2 passes (**b**,**f**), 3 passes (**c**,**g**), 4 passes (**d**,**h**).

**Figure 8 materials-13-04932-f008:**
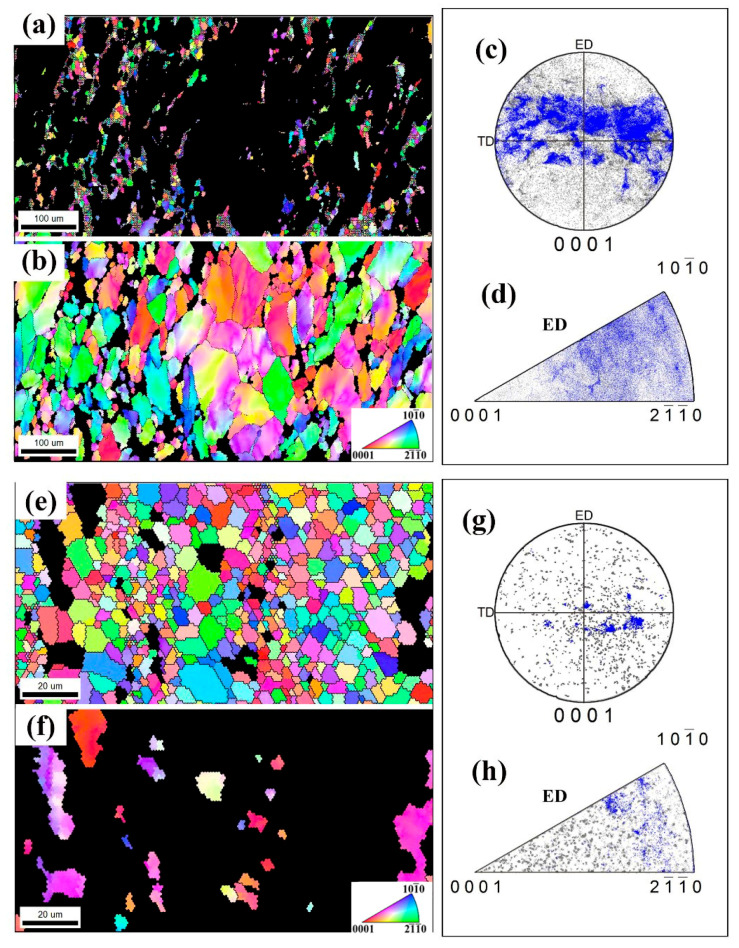
Inverse pole figure (IPF) coloring maps of DRXed (**a**,**e**) and unDRXed (**b**,**f**) grains after one-pass (**a**,**b**) and three-passes (**e**,**f**) deformation, and corresponding (0001) PF (**c**,**g**) and IPF (**d**,**h**).

**Figure 9 materials-13-04932-f009:**
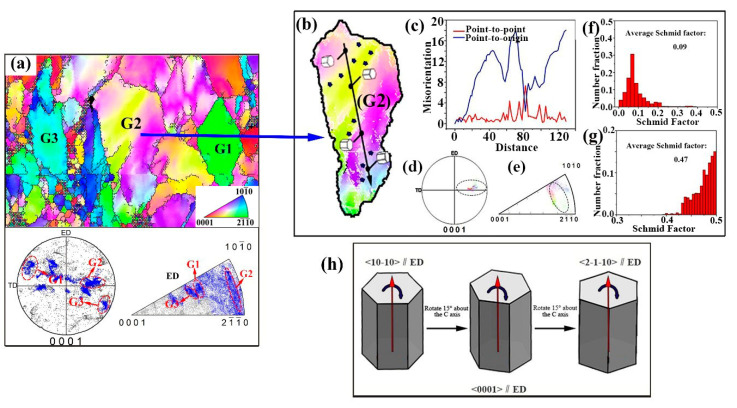
IPF coloring map and corresponding (0001) PF and IPF after 1 pass (**a**). The OIM diagram of grain G2 (**b**) and the corresponding line graph of misorientation angle along the AB (**c**), PF (**d**), IPF (**e**), Schmid factor (SF) graph of basal plane (**f**), SF graph of prismatic plane (**g**) in (**b**); schematic diagram of prismatic slip (**h**).

**Figure 10 materials-13-04932-f010:**
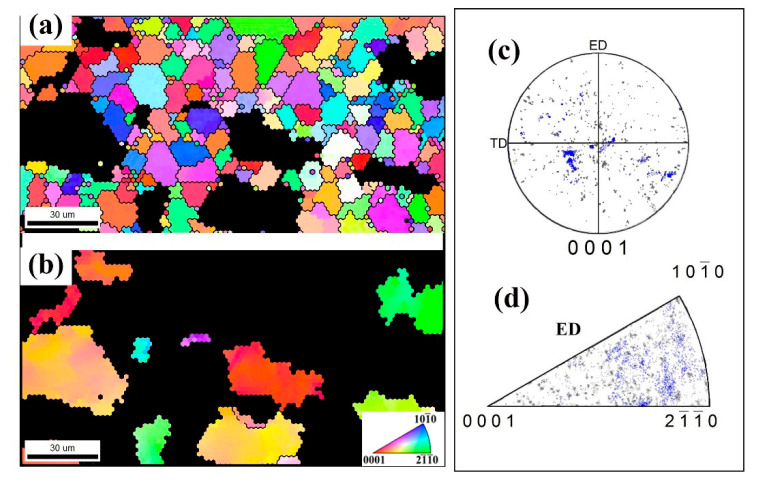
IPF coloring maps of DRXed (**a**) and unDRXed (**b**) grains after two-passes, and corresponding (0001) PF (**c**) and IPF (**d**).

**Figure 11 materials-13-04932-f011:**
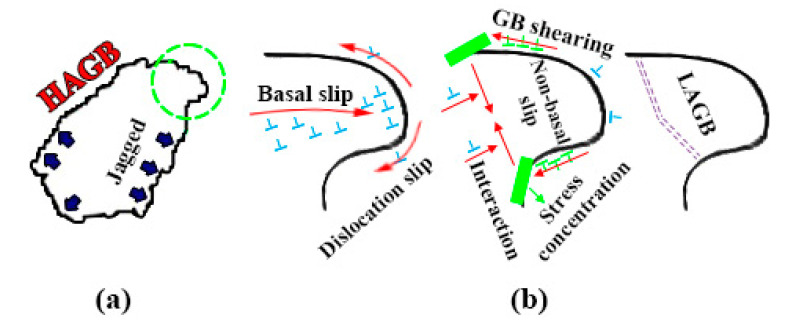
Schematic diagram of discontinuous dynamic recrystallization (DDRX) process. (**a**) Schematic diagram of a typical grain where a DDRX process is taking place; (**b**) Schematic diagram of DDRX process.

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
