# Peer review of "Microstructure and Texture Evolution of Mg-Gd-Y-Zr Alloy during Reciprocating Upsetting-Extrusion"

_materials, 2020, doi:10.3390/ma13214932_

Round 1

Reviewer 1 Report

Introduction is very generic and is some cases it is poorly connected with the aim of the proposed paper. Is there any novelty in the of the proposed investigations?

Please rewrite Abstract. Including purpose of research, research method and results.

Please check equation (1)

experiment conditions of upsetting and extrusion process effect microstructures. Please comment on conditions of each process such as force (or pressure), friction coefficient, velocity, and others.

Please revise first image in Figure 1. There is no gap between die cavity and billet. I think diameter of die cavity in first image is the same that of upsetting sample.

Please indicate dimensions of the die. Diameter of die cavity is important factor.

Please explain the RUE1, RUE2, RUE3, RUE4. What is the difference?

Please present samples about upsetting and extrusion. indicate dimensions to sample.

Please present images of experiment equipment for Figure 1.

Please change microstructures in Figure 2(a) and all of the Figure 3 to high-resolution.

Please also present low magnification microstructures about each RUE process in Figure 3.

is the RUE1, RUE2, RUE3, RUE4 same 1 pass, 2 passes, 3 passes, 4 passes? If so, please use same expression.

Please check word spacing for example leave a space between number and unit.

I think the discussion chapter is unnecessary. Indeed, this chapter present a summary of data present in chapter of results.

For manuscript improvements a new conclusion draws should be made.

Reviewer 2 Report

The manuscript “Microstructure and texture evolution of Mg-Gd-Y-Zr alloy during Reciprocating Upsetting-Extrusion” presents findings obtained from the investigation of that forming technique using microscopy and crystallography-based analysis techniques. The manuscript will be reconsidered appropriate for publication in Materials only after major revision. The authors should revise their manuscript according to the following remarks.

Figure 2b: please correct, the word

Line 89: “XRD..phase evolution”, perhaps I missed it in the text but I did not noticed that XRD study revealed any information regarding phase transition. If that is the case, please correct to “identification”.

Line 92: “etc.” please remove and write explicitly what was studied and using what technique.

Line 147: Overall, what were criteria and fundamental assumptions to separate DRXed grains from originally grains? This should be addressed in the manuscript.

Figure 6: Please add the following information – direction of the IPFs.

Line 152: the authors suggest that due to deformation temperature, no twinning occur. I would like to ask if the authors examined the microstructure prior to heat treatment, especially at the first cycle. Twinning might be present at coarse-sized grains and thermal treatment will enable their growth. This can be appreciated from the analysis of texture.

Line 189: “the formation of DRX grain consumes a lot of dislocations “, please re-write this sentence!!

Line 200: The active mechanism is not clear, by suggesting that LAGBs are “absorbed”, do the authors suggest dislocation inhalation at sinks? What about entanglement resultant of particles?

Line 207: authors’ generalized statement is not correct! There are many cases where homogenization will lead to growth of nucleated twins, thus yield preferential texture. Please correct.

Line 214: a proper description of texture components is missing in the text. This should be added, with special attention to the evolved texture following this forming procedure.

Line 233: It can be found?! please rewrite this sentence.

Line 240: “is a kind of “, please correct. Also, RUE is not a “deformation method”, no such thing as deformation method (!). Instead, this is a technique used in forming of metals.

line 246-247: please rewrite this sentence.

Line 249-250: what was the conceptual idea behind this segregation (60-90deg for example). Although twins were not observed, why didn’t the authors discard or gave attention to twins boundary (86±5, 56±5..)?

Line 257: There should be some correlation between texture transition and/r texture modification, to defects (dislocation density – entanglement, pile up..).  This should be included in the manuscript, describing how microstructure and texture are related throughout the forming procedure. This should be explained by macro and micro-texture yet not only by the analysis of a single grain that not necessarily represent the material or deformation mechanism.

261: Figure 8 does not represent grain orientations, it describes deviation from a preferential direction (in this case, c-axis to TD).  As the authors decided not to include grain boundaries, it is not possible to distinguish neighboring grains with similar orientation. Please correct.

Overall, I do not see any benefit of showing this image (Fig. 8). Instead, the authors should show ODF and discuss variation of texture components. As an alternative, they can further explain this point quantitively through f(g) description of the 0002 PFs near the TD.

Also, and very important - the authors should address the issue of texture statistics – as an example, texture data extracted from Fig.8 is based on the scanning of a small region. It is well known that the strain flow and fiction alter microstructure and texture at different location in the sample. How did the authors consider this point in their research? I wonder if at different area, the trend of texture is similar, probably not (see for example, top plane compared to center).

Line 268: Texture component is a term which was not use/described properly in this article. Please correct!

Line 270: how did the authors calculate the stress matrix used for Schmidt factor determination? Why wasn’t it described in the text? Also, did the surrounding grains for the G2 and G3 displayed similar orientations? Also, what was the grain boundary character between these grains (G2/g3) and their surrounding? If they are not similar, there should not be a base for comparison or discussion! Please correct accordingly.

Line 273: The authors suggested Schmidt Factor but did not described what is the barrier – is it slip system (which one? Prismatic, basal...) or twining (tensile, compressive, de-twining... ?)

Lines 278 to 287: a reparative idea which was basically described in lines 272 to 277. Please correct.  

Line 289: the Schmid factor histogram is based on what image? No caption description of (f) and (g). The manuscript should not be sent for review in current state. PLEASE CORRECT!.

Line 292: Please change grain number following the hierarchical description of the individual grains (G1, G2, and G3..).

Line 294: Authors should describe in the relevant text that disorientation angle calculation is from origin point to each point along that line (‘AB’).

Line 291: The authors try to explain DRX behavior through the examination of a single grain? again, I do not understand what was the criteria and assumptions behind this paragraph, this idea was already discussed earlier in this manuscript. Please explain and give scientific justification to that paragraph.

Line 296: pole figurer?! The manuscript should not be sent for review in current state.

Line 270: why not G1 and G2??

Line 307: the idea behind LAGBs to HAGBs transformation was already described in text relevant to Fig.6.

Line 310: “likely to be the main deformation mechanism in grains”… for a specific grain orientation and loading axis. Please correct.

Line 321: “texture intensity”, to what texture component do the authors point at?? Perhaps they aimed to the M.R.D ? PLEASE CORRECT!

Line 332: “HAGB has a high dislocation density”, it is not possible! The authors meant grains and not grain boundaries. This entire sentence needs to be rephrased.

Reviewer 3 Report

The current study investigates the texture evolution and microstructure of Mg-Gd-Y-Zr alloy processed by reciprocating upsetting-extrusion.

The manuscript is well organized and interesting results are presented. However, some minor issues need to be considered as follows:

  • The introduction section should be extended to include more details about the current problem statement and the direct applications to the processed alloy. 
  • The abstract should start with one or two sentences that describe the importance of the current work to attract the reader.
  • The conclusion section should focus on the main contribution and novelty of the current study along with the presented summary of the main results.    

Round 2

Reviewer 1 Report

I check revised manuscript

Reviewer 2 Report

All comments were adequately addressed.

All the best

Reviewer 3 Report

The review comments and recommendations are well addressed by the authors.